# Robust, Superhydrophobic Aluminum Fins with Excellent Mechanical Durability and Self-Cleaning Ability

**DOI:** 10.3390/mi14030704

**Published:** 2023-03-22

**Authors:** Wenbo Su, Xiangyou Lu, Yunxiang Shu, Xianshuang Liu, Wen Gao, Jianjie Yao, Zhuang Niu, Yuanlai Xie

**Affiliations:** 1School of Environmental and Energy Engineering, Anhui University of Architecture, Hefei 230601, China; 2Institute of Plasma Physics, Hefei Institute of Materials Science, Chinese Academy of Sciences, Hefei 230031, China

**Keywords:** self-cleaning, aluminum, femtosecond laser, durability, finned heat exchanger, superhydrophobicity

## Abstract

The self-cleaning ability of superhydrophobic metal surfaces has attracted extensive attention. The preparation of superhydrophobic material using the coating method is a common processing method. In this experiment, aluminum fins were processed by laser etching and perfluorinated two-step coating. The aluminum surface was modified using a femtosecond laser and 1H,1H,2H,2H- perfluorooctane triethoxysilane (PFOTES). A superhydrophobic aluminum surface with excellent mechanical stability and self-cleaning properties was obtained with the superhydrophobic contact angle (WCA) of 152.8° and the rolling angle (SA) of 0.6°. The results show that the superhydrophobic surface has an excellent cleaning effect compared with an ordinary surface in unit time. Then, a wear resistance test of the superhydrophobic surface was carried out by using the physical wear method. The results show that physical wear had a low influence on the hydrophobic property of the specimen surface. Finally, the Vickers hardness analysis found that the superhydrophobic surface hardness was significantly better than the ordinary surface hardness compared with the superhydrophobic surface hardness. Based on the excellent self-cleaning properties, wear resistance, and robustness of superhydrophobic materials, the laser-etched and perfluorinated superhydrophobic aluminum fins designed and manufactured in this study have broad application prospects in improving the heat transfer efficiency of finned heat exchangers.

## 1. Introduction

Finned heat exchangers are widely used in industrial processes and energy conversion processes, such as heating, ventilation, air conditioning, and refrigeration, due to their compactness and low cost [1,2,3]. Finned heat exchangers generally achieve heat exchange through fluid transfer and improve heat transfer efficiency by utilizing the efficient surface area of the fins [4]. However, during daily use, dust particles in the environment can accumulate on the surface of the heat exchanger fins after long periods of operation. Dust particles are easily attracted to water molecules in the air, leading to the formation of a wet dust layer, which can gradually form a stubborn layer of dirt after a long period of accumulation [5,6]. The formation of a stubborn dirt layer and scale on the heat transfer surface has a serious impact on the performance of the thermal fluid and the overall efficiency of the heat exchanger. A scale layer on the heat transfer surface can lead to blockage of the fluid flow, increase the air measured wind resistance, and reduce the thermal conductivity of the fins, resulting in a reduction in the thermal performance of the heat exchanger. To avoid the degradation of heat exchanger performance, the fin surface dust needs to be removed promptly and effectively [7,8,9]. Therefore, an efficient, economical, and convenient dust removal method is required.

Superhydrophobic surfaces have properties such as anti-frost [10], anti-corrosion [11], self-cleaning [12], anti-icing [13], fluid drag reduction [14], and enhanced condensation heat transfer [15]. Due to their extremely low surface energy [16], superhydrophobic surfaces have great potential for mitigating dust deposition [17]. Zhang et al. used superhydrophobic coatings for droplet dust removal, which in turn improved photovoltaic efficiency, and found that the photovoltaic efficiency could be increased by 20% per MW of PV plant [18]. Wu et al. investigated the dynamic process of dust removal during condensation on superhydrophobic surfaces and found that the superhydrophobic surface had the best dust removal effect [19]. Yang et al. investigated the dust removal effect of superhydrophobic copper surfaces in hot and humid environments and found that the dust coverage of superhydrophobic surfaces decreased by 69.9%, 61.1%, and 3.59% compared to normal, hydrophilic, and hydrophobic surfaces, respectively [20]. Liu et al. studied superhydrophobic aluminum surfaces prepared from aluminum alloys of different purity and found that the higher the purity of the aluminum surface, the better the superhydrophobicity and the better the self-cleaning performance; however, the wear resistance and hardness would become lower [21], which is not conducive to practical industrial applications. Currently, the main methods for preparing superhydrophobic metal surfaces include laser processing [22], chemical etching [23], anodic oxidation [24], and electrostatic spraying [25]. Sun et al. used sandblasting and chemical modification to obtain layered structures and low surface energy, and the superhydrophobic aluminum surfaces exhibited excellent thermal stability and self-cleaning effects [26]. Tong et al. used femtosecond lasers to adjust the texture of aluminum surfaces dimple patterns, and due to the presence of surface hardening on the laser-patterned surface, a very mechanically stable superhydrophobic aluminum surface was obtained [27]. Sataeva et al. used laser processing and chemisorption of fluoxysilanes to impart good water and ice resistance to the material, and the interaction of laser processing and fluoxysilanes was able to increase the rate of superhydrophobic coatings in a controlled manner compared to other methods, with high mechanical and chemical resistance [28]. Wang et al. developed a novel strategy to fabricate mushroom-like structures in situ constructed by nanowhisker–nanowire hierarchical architectures on copper mesh through a more flexible bottom-up method. The final morphology is highly controlled by adjusting corresponding synthetic conditions. The superhydrophobic copper mesh with a static contact angle of 151.8° and a sliding angle of 3.6° possesses superior anti-fouling and self-cleaning performance related to the pollution of daily drinks and dirt [29]. However, the superhydrophobic surfaces prepared by the usual methods can have some inherent drawbacks that are not conducive to practical applications. For example, the mechanical resistance of the metal surface after laser processing is low and the hardness can be reduced [30]. The superhydrophobic coatings obtained by electrostatic spraying become less wear-resistant and the coatings become prone to wear [31]. Chemical etching methods can suffer considerable material losses and the process is too complex to be used on a large scale [32]. Laser processing in conjunction with chemisorption is therefore an effective means of providing mechanical properties on metal surfaces [14].

In this study, a femtosecond laser was used to process the micro- and nanostructures of aluminum surfaces, and 1H,1H,2H,2H-perfluorooctyltriethoxysilane (PFOTES) was used to chemisorb against aluminum surfaces to form combined surfaces, aiming to obtain superhydrophobic materials with robust, durable, and mass-usable properties. Dust-removal performance tests were carried out to observe and analyze the self-cleaning effect of different surfaces on dust, and in conjunction with the physical admissibility analysis of the particles, a link was established between the driving force required for the sliding of the particles and the contact angle of the metal surface, which in turn provided theoretical support for the self-cleaning performance of the superhydrophobic material. The Vickers hardness test results showed that the surface hardness of the metal surface was significantly enhanced after the chemisorption of fluorxysilane. Next, the wear resistance of the material surface was studied, and the results showed that the metal surface exhibited good mechanical robustness due to the solid protection provided by the superhydrophobic coating. The experimental analysis shows that the ability to achieve high-speed controlled laser processing of aluminum surfaces to create robust, highly mechanical, and self-cleaning superhydrophobic coatings is of great significance for finned heat exchangers to achieve high efficiency and cost-effective heat transfer and has a wide range of applications in refrigeration and heat exchange engineering and in daily use.

## 2. Experimental Materials and Methods

### 2.1. Experimental Materials

Anhydrous ethanol (99.99%), hydrochloric acid (HCL, 37%), 1H,1H,2H,2H-perfluorooctyltriethoxysilane (PFOTES, ≥98%), and acetone (≥99.9%) were obtained from Shanghai Aladdin Biochemical Technology Co. (Shanghai, China). All chemicals are analytically pure reagents and can be used without additional processing. The content of 1A99 aluminum alloy is shown in Table 1.

### 2.2. Preparation of Superhydrophobic Surfaces

In this experiment, a high-purity aluminum sheet with a mass fraction of 99.9% was selected, and a smooth aluminum sheet of 10 mm × 10 mm × 1.5 mm was taken and polished with metallographic sandpaper of 600, 800, 1200, and 2000 in turn, until the surface was smooth without obvious scratches. After surface polishing, there may be some small particles on the surface of the sample. To reduce the influence of particles on the experiment, we placed the sample in anhydrous ethanol solution, carried out ultrasonic cleaning for 10 min, and then put the sample into a dryer for drying to obtain sample 1#. The microstructure of the aluminum surface was prepared by a FemtoYL-IR-40W femtosecond laser. The laser wavelength was 1035 nm, the laser pulse width was 400 fs, the output frequency was set to 400 kHz, and the laser scanning rate was 400 mm/s for 20 times, as shown in the figure. The surface of the sample was treated in a range of 10 mm× 10 mm and the scanning interval was 92 μm Laser processing was carried out in the air. The processing equipment and route are shown in Figure 1. Immediately after laser treatment, the specimen was ultrasonic cleaned in anhydrous ethanol and deionized water for 10 min, then removed and dried naturally to obtain sample 2#. The surfaces of the femtosecond-laser-melted samples were pretreated first. The samples were placed in anhydrous ethanol, cleaned for 2 min with the ultrasonic cleaning instrument, and then dried. The cleaned samples were immersed in the perfluorooctane triethoxysilane ethanol solution for 1H,1H, 2h,2H- for 2 h, and then dried in a drying oven at 100 °C for 30 min. Superhydrophobic aluminum sample 3# with low surface energy was obtained.

### 2.3. Superhydrophobic Aluminum-Based Surface Performance Characteristics

The contact angles of the samples were tested at room temperature using a contact angle meter (DSA100, KruSS, Hamburg, Germany), which in turn allowed observation of the micro-nanostructure on the surface of the specimens, using an S-4800 cold field emission scanning electron microscope (SEM; Zeiss SIGMA HD, Jena, Germany) to observe the micro-nanostructure on the specimens, using a medium of deionized water with a droplet volume of 2 uL. Five different locations were selected and averaged. Next, the hardness of the specimens was measured using a microhardness tester (Wilson Hardness 401MVD, Wilson, Chicago, IL, USA) to measure the microhardness of the specimen at five different locations and the average value was taken as the hardness value of the specimen. Finally, the WCA and SA of the specimens were measured by applying pressure to the specimens with a 200 g weight and moving them over a 1000-mesh sandpaper with a 15 cm cycle, for a total of 10 cycles, to reflect the strength of their durability properties by the change in WCA and SA.

### 2.4. Dust Removal Test

The experimental system consisted of a dust cleaning system, a microscope camera system, and a data acquisition system. The dust cleaning system consisted of a dusting unit and a spraying unit. The dust unit was used to simulate the process of dust deposition on the fin surface and the spray unit was designed to investigate the cleaning performance of the dust caused by condensation. We fixed the specimen on a slope inclined at an angle of 10° and the dust particles were blown by a variable-frequency fan to blow the dust, which was then deposited uniformly on the surface of the specimen by gravity. After waiting for 20 min, a dust-covered layer was formed on the surface. The dust-covered specimen was then placed in the condensate formation unit. The condensate formation unit consisted of a nano-humidifier, insulating cotton, a beaker, and the specimen. The nano-humidifier provided air humidity, and the insulating wool was arranged around the test piece to avoid condensation formation around the test piece. The beaker was used to collect condensation droplets from the surface of the test specimen. The microscope camera system consisted of a metallographic microscope and a CCD camera. The metallographic microscope was used to observe and record the process of condensation formation on the surface of the test specimen and the CCD camera was used to observe and record the process of adsorption and cleaning of dust on the surface of the test specimen by water droplets sliding off after condensation formed on one side of the specimen. The data acquisition system used an electronic balance to measure and record the weight of the remaining dust on the surface of the specimen at different surface wettability. The experiments were carried out on 1# plain aluminum, 2# laser-etched aluminum, and 3# laser-etched + fluoride-treated aluminum surfaces, and the dust removal was observed and analyzed on these three specimens.

## 3. Results and Discussion

### 3.1. Surface Micro-Nanostructures

Femtosecond lasers are seen as a versatile method for the processing of micro- and nanostructures [33]. In general, the structure of the chemical composition at low surface energies determines whether a substance is hydrophobic, while the microscopic roughness of the substance surface determines the hydrophobic strength [34]. The superhydrophobic aluminum surface structure was analyzed using SEM for detailed observation. As shown in Figure 2a–c, the surface of untreated specimen 1# showed a relatively flat and smooth surface structure. Specimen 1# was obtained as specimen 2# after femtosecond laser processing, as shown in Figure 2d–f. The laser beam was focused onto the sample surface through the objective lens. At this point, the temperature of the processed area rose rapidly due to the high energy density effect of the laser. The specimen vaporization pressure grew relatively slowly, and the equilibrium of the transformation was broken. The surface of the specimen was destroyed by high-temperature etching and new micro-nano-periodic structures were gradually formed. The microgrooves produced by laser ablation were extensively covered by irregular nanostructures. As shown in Figure 2d, gratings with a spacing of 5–10 μm were formed on the metal surface and distributed in a periodic parallel pattern. As shown in Figure 2e,f, the surface of the metal forms spore-like processes and dents, showing a papillary structure, with the ridges and valleys of the microgrooves having an abundance of irregular micro- and nanostructures, the presence of which increases the surface roughness of the material and the density of the micro- and nanostructured regions.

On this basis, the surface of the micro-nanostructure was fluorinated to reduce the surface energy of the metal and obtain specimen 3#. As shown in Figure 2g–i, it can be observed that after the fluorination treatment, a nanowire-like structure was formed, its surface roughness became larger, and its hydrophobicity became stronger. By comparing the surface structure of 2# and 3#, it was found that the micro-fine raised structure on the surface of 3# after fluorination treatment was corroded away by hydrochloric acid, the surface complexity was reduced by the large number of nanoflakes present on the aluminum surface after laser etching, and part of the nanorods were fused together to obtain a better surface micro-fine structure. The presence of micropores and nanoflakes helped to obtain a rougher surface, which, in interaction with the low free energy obtained during fluorination treatment, enables a better superhydrophobic effect to be obtained. The micro- and nanostructured superhydrophobic surfaces possess better mechanical stability and wear resistance compared to those prepared by other methods [30]. Compared to chemical methods of preparing superhydrophobic surfaces, the preparation method becomes more environmentally friendly and greener, as well as safer.

### 3.2. Surface Wettability Analysis

Wettability is one of the important characteristics of solid surfaces and is related to surface microstructure and surface energy [35]. In general, the contact angle (WCA) is used to characterize the wettability of a solid surface. Depending on the WCA, solid surfaces can be classified as hydrophilic (CA < 90°), hydrophobic (90° < WCA < 150°), and superhydrophobic (CA > 150°) [36]. Specimen 2#, after femtosecond laser treatment, did not possess superhydrophobicity immediately; the laser constructed a micro-nanostructure on the metallic aluminum surface and water droplets were able to spread completely on the alloy surface, exhibiting a superhydrophobic state. After leaving the specimens in an air environment for a period, a weak chemical reaction occurred on the surface of the specimen, oxidation forms on the surface, and the specimen can eventually reach a superhydrophobic state. The contact angles of three sets of specimens were measured in an atmospheric environment and the results are shown in Figure 3 below. The contact angle of the untreated aluminum metal surface to water was close to 90°. After femtosecond laser etching, the contact angle to water increased to around 130° and the surface became hydrophobic. According to Young’s theory, the WCA of an ideal smooth surface depends only on the surface energy and is independent of the surface structure [37]. The laser-treated specimens were then fluorinated to reduce their surface energy and the contact angle to water reached about 150°, achieving a superhydrophobic effect.

### 3.3. Dust Removal Experiments

It is well known that most organisms that exhibit superhydrophobic properties also have self-cleaning properties, and self-cleaning is one of the important properties of superhydrophobic surfaces [38]. This works because the adhesion of dust particles to water droplets is much greater than to the superhydrophobic surface, and when the water droplets roll off the superhydrophobic surface, the dust particles are carried away from the superhydrophobic surface together with the water droplets [19]. Thus, superhydrophobic surfaces are capable of self-cleaning. We examined the self-cleaning ability of different surfaces through spray experiments. As shown in Figure 4 below, a sample was fixed on an inclined slope with an inclination angle of 10° and placed in a sealed container. Under natural gravity, a uniform layer of coal dust (particle size: 20–40 μm) was applied to the surface of the sample to simulate ‘dust’. A nano-humidifier was also used to increase the humidity of the air. Figure 4e shows the distribution of dust on the surface of each of the three samples after 2 h. It is easy to see that the cleaning effect of the aluminum-based surfaces varied considerably after the condensation. The dust on the surface of specimen 1# was no longer dispersed by the action of the condensate but was closely packed together by the action of the water molecules to form an accumulation. The boundaries of these particles became blurred, and the particle forms were indistinguishable, forming a layer of powder fouling. Specimen 2# showed a certain cleaning effect as the dirt layer formed a thin film of dust that adhered to the surface of the specimen under the effect of hydrophobicity and did not form a thick layer of dirt compared to the normal aluminum-based surface. In terms of specimen 3#, under the action of surface superhydrophobicity and its own gravity, the superhydrophobic aluminum-based surface dust basically achieved the cleaning purpose.

The dust removal behavior that occurs on superhydrophobic surfaces is the result of competition between adhesion and driving forces. We have learned that a certain self-cleaning effect can be achieved using a certain tilting angle alone in the absence of water [19]. Considered from the aspect of physical adhesion, the adhesion force is the main reason why dust particles adhere to solid surfaces. Adhesion forces refer not to physical macroscopic angular forces but to the resistance caused by friction and contact angle hysteresis between the dust/condensate and the contact surface, which includes its own gravity, van der Waals forces, capillary forces, electrostatic forces, etc. [39]. In the self-cleaning process, the above-mentioned self-cleaning pattern occurs once the queuing condition is satisfied, as shown in Figure 4a–d, where the condensate mixes with the dust and finally the self-cleaning process occurs when the queuing condition is satisfied. Figure 4f shows the force analysis of a dust-laden sloshing droplet.

#### 3.3.1. Gravity on Dust-Containing Droplets

The force of gravity on a droplet on an inclined plane is:(1)$$G=Mg=\rho gV=\frac{4\pi \rho gR^{3}}{3}$$
where: *G* is the gravity of the droplet particle itself; *ρ* is the density of the dust-laden droplet particle; *R* is the equivalent radius of the droplet; *M* is the weight of the droplet itself; and *g* is the acceleration of gravity.

The component forces in the direction of the parallel solid surface are:(2)$$G_{1}=G\sin\;\alpha =\rho gV\sin\;\alpha =\frac{4\pi \rho gR^{3}}{3}\sin\;\alpha$$

The component forces perpendicular to the surface of the solid are:(3)$$G_{2}=G\cos\;\alpha =\rho gV\cos\;\alpha =\frac{4\pi \rho gR^{3}}{3}\cos\;\alpha$$
where: *G*_1_ is the component force in the direction parallel to the solid surface; *G*_2_ is the component force in the direction perpendicular to the solid surface; and *α* is the angle of inclination of the solid surface.

#### 3.3.2. Capillary Forces on Dust-Containing Droplets [40]

The capillary force on the droplet is:(4)$$F_{c}=4\pi R\gamma \cos\;\theta$$
where: *F_c_* is the capillary force on the dust-containing droplet particle; *γ* is the surface tension of the liquid; and *θ* is the surface tension of the dust-containing droplet particle.

#### 3.3.3. Van der Waals Forces on Dust-Containing Droplets [41]

The van der Waals force on the droplet is:(5)$$\begin{array}{ll} F_{vdw} & =\frac{AR}{6z^{2}}*\left[\frac{1}{\left(1+\frac{R}{1.48rms}\right)}+\frac{1}{{\left(1+\frac{1.48rms}{z}\right)}^{2}}\right]\\ & =\frac{AR}{6z^{2}}*\left[\frac{{\left(1.48rms\right)}^{3}{+2z\left(1.48rms\right)}^{2}+Rz^{2}+2.96rmsz^{2}}{\left(1.48rms+R\right){\left(1.48rms+z\right)}^{2}}\right] \end{array}$$
where: *A* is Hamaker’s constant, which reflects the magnitude of the strength of the van der Waals force and is related to the surface energy between the contacting solids; *z* is the vertical distance between the particle and the plane of contact; and *rms* is the root mean square roughness of the solid surface.
(6)$$A=24\pi z^{2}\sqrt{\gamma_{p}\gamma_{s}}$$

Substituting (6) into (5) and collating gives:(7)$$F_{vdw}=4\pi R\sqrt{\gamma_{p}\gamma_{s}}*\left[\frac{{\left(1.48rms\right)}^{3}{+2z\left(1.48rms\right)}^{2}+Rz^{2}+2.96rmsz^{2}}{\left(1.48rms+R\right){\left(1.48rms+z\right)}^{2}}\right]$$
where: *γ_s_* is the surface tension of the solid wall and *γ_p_* is the wetting contact angle of the dust-containing droplet particles.

If the droplet is to complete its automatic slide, it needs to overcome the adhesion force *F* of the solid surface to the droplet particle, which is equal to the support force *F_n_* of the solid surface to the droplet particle, as shown by Newton’s three laws, i.e.:(8)$$F=F_{n}=G_{2}+F_{vdw}+F_{c}$$
where: *F* is the droplet particle surface adhesion force and *F_n_* is the support force of the solid surface on the droplet particle.

It is also known that the static frictional force *F_f_* along the horizontal direction is equal to the component force of the particle along the surface parallel to the solid, i.e.:(9)$$F_{f}=G_{1}$$
where: *F_f_* is the static frictional force of the droplet particles.

In general, capillary forces are seen as the predominant adhesion force between particles and solid surfaces under moist environmental conditions [42]. This leads us to focus on Equation (4), where the value of the adhesion force is smaller for particles with approximately larger wetted contact angles, with a superhydrophobic surface WCA of 152.8° and a normal surface WCA of 85.96°. Thus, the capillary force developed on a superhydrophobic surface is smaller than on a normal surface. In addition, the van der Waals force is only a small part of the force in a humid environment. As the root mean square roughness of a superhydrophobic surface is greater than that of an ordinary surface, however, its free energy is much less than that of an ordinary surface, and as can be seen from Equation (7), the van der Waals force on a droplet particle on a superhydrophobic surface is also much less than that on an ordinary surface.

Figure 5 below shows the residual dust weights of the different specimen surfaces. After a 2 h dust removal process, the residual dust weight on the hydrophobic and superhydrophobic aluminum surfaces was much less than the residual dust weight on the normal aluminum surface in this study, with the smallest residual dust weight on the superhydrophobic aluminum surface. The residual dust weight of the superhydrophobic aluminum surface was 50% lower than that of the hydrophobic surface and 70% lower than that of the normal surface. The reason for this phenomenon is that dust particles on hydrophobic and superhydrophobic aluminum surfaces are separated from the metal surface by water droplets to achieve a cleaning effect, and dust particles on normal aluminum surfaces are removed only by absorbing condensate.

### 3.4. Vickers Hardness Analysis

Laser treatment has a significant impact on the microhardness of the sample surface. In practice, the fragile micro- and nanostructures of superhydrophobic surfaces will seriously affect their service life [31]. Therefore, improving the mechanical durability of the material itself has become a key factor in enabling its widespread use [43]. We used a square conical diamond with a relative face angle of 136° to press into the surface of the material, held it for a specified time, and then used the measured diagonal length of the indentation to calculate the magnitude of the hardness according to the equation. As can be seen in Figure 6a below, the specimens showed a papillary, raised structure on the surface after laser treatment and therefore the hardness measurement after laser treatment may be influenced by the measurement area. The three specimens were analyzed separately for microhardness, and to ensure the accuracy of the testing process, the hardness values were selected for five different locations on the specimen surface. The results are shown in Figure 6c. The surface microhardness of specimens 1# to 3# were (37.7 ± 0.8) HV_0.2_, (27.6 ± 1.2) HV_0.2_, and (58.8 ± 10.3) HV_0.2_. After laser treatment, although the formation of recast layers, such as small metal particles on the surface of the material and the refinement of the particles, can produce more grain boundaries and make crystal dislocation slippage more difficult, after laser treatment the surface of the specimen loses some structural protection afterwards, i.e., the microhardness is slightly reduced. After the fluorination treatment, the surface wear resistance of the material increased and the milky spore complexity of the surface was slightly flatter than in specimen 2#, so that conical diamond dislocations on the material surface became more difficult, giving the material surface a layer of “armor” and increasing the hardness of 3# compared to 1# and 2#. However, because of the presence of the milky projections and concave structures on the surface of the material, the hardness could reach 66 HV_0.2_ at the projections, but only about 47 HV_0.2_ at the flattening. For #2 and #3, the specimens also showed a micro-groove structure on the surface, but the material underwent a multiplicative increase in hardness before and after the fluorination treatment. Specimens that were laser-etched are prone to surface displacement, resulting in a reduction in their hardness. However, after the fluorination treatment, with the increase in the roughness and wear resistance of the material itself, we found that the hardness of the material underwent a dramatic increase. With the laser- and fluorination-treated specimens treated in this paper showing a significant increase in hardness compared to the untreated specimens.

### 3.5. Surface Stability and Durability Analysis of Specimens

Poor mechanical robustness greatly limits the scale of superhydrophobic aluminum surfaces in practical applications, where even slight touch or abrasive forces can damage the surface structure and thus lose hydrophobicity [30]. Therefore, we investigated the mechanical robustness of the specimens after laser etching + fluorination treatment by abrasion experiments. As shown in Figure 7 below, the superhydrophobic specimens were rubbed in one direction under a load of 200 g, extended horizontally on 1000 mesh sandpaper with a row spacing of 15 cm, and the CA of specimen 3# showed a decrease with increasing distance of sandpaper wear, but only within a small range of variation (<10°) of decrease, while the SA kept increasing with increasing wear. After the specimens were rubbed back and forth on sandpaper for 150 cm, they lost their superhydrophobicity, but successfully maintained some hydrophobicity. Compared with other similar superhydrophobic aluminum surfaces, the durability of the superhydrophobic aluminum surfaces designed and prepared in this paper was significantly enhanced.

### 3.6. Surface XPS Analysis

It is well known that the surface chemistry of the substrate is one of the most important factors affecting wettability [36]; therefore, we performed XPS analysis of the specimens to study the surface chemistry and state under the combined effect of laser and fluorination treatments, and the results are shown in Figure 8. It was observed that there were no peaks of F and Si elements on the surface of normal aluminum as well as on the laser-etched aluminum surface (see Figure 8a,b). The presence of the F1s peak in Figure 8c represents the effective fluorination of 1H,1H,2H,2H-perfluorooctyltriethoxysilane on the surface of the specimen. The successful grafting of the fluorine element is attributed to the presence of the -OH group on the surface of the specimen and the minor contribution of the silicon atom can be deduced from the observation of the Si2p peak present in Figure 8c below. Micro/nanostructures with low surface energy tend to produce desirable superhydrophobicity [34]; therefore, the optimal superhydrophobicity of fluorinated laser-etched aluminum-based specimens is the result of a combination of micro/nanostructure and degree of modification.

The XPS spectrum of specimen 3#, represented in Figure 8c, supports the effective grafting of the perfluorooctane triethoxy-silane molecule onto the etched aluminum surface. The XPS analysis further investigated the elemental composition and chemical valence of the surface of 3#. As shown in Figure 8d below, the binding energies of the four peaks in the high-resolution spectrum of C1s were 293.24 eV, 290.75 eV, 287.87 eV, and 284.8 eV, corresponding to C-F, O-C=O, C-O, and C-C/C-H, respectively (mainly due to air contamination). In addition, the peak in O1s at 531.76 eV can be attributed to the absorption of water and hydroxide on the Al surface, forming Al-O (Figure 8e). As can be seen in Figure 8f, the hidden peak of the main F1s peak mainly reflects the presence of C-F and F-, whose binding energies are located at 687.63 eV and 684.64 eV, respectively, where the C-F bond has excellent stability against humid atmospheric action, which is key to maintaining the excellent low surface energy and superhydrophobicity of the specimen [44].

## 4. Conclusions

An aluminum-based superhydrophobic surface with good mechanical stability was prepared by femtosecond laser treatment and fluorination. The surface morphology and chemical composition of the samples were studied using SEM and XPS measurements, and the structure of the micro-pit array and the hydrophobic groups successfully grafted were observed. In addition, the experimental results show that the hardness of the superhydrophobic surface prepared by this method is about 1.5 times higher than that of the ordinary surface. The aluminum surface maintained a high WCA after being worn with 150 cm sandpaper. In addition, compared with the dust removal effect of an ordinary surface and superhydrophobic surface, there is no dirt layer on the surface of superhydrophobic aluminum. Through physical force analysis, the adhesive force of liquid droplet particles is related to the surface energy and contact angle of metal in the process of dust removal. It was found that the better the superhydrophobicity of the metal surface, the lower the adhesive force of the solid surface to the dust coverage of liquid particles. This provides a strong theoretical support for the excellent self-cleaning ability of a superhydrophobic surface. The experimental data show that the dust weight of the superhydrophobic aluminum surface is only about 30% of the normal surface. The experimental results show that the superhydrophobic aluminum surface prepared in this paper has excellent self-cleaning ability and excellent durability and mechanical properties. With low cost and high efficiency, this method has broad application prospects in the field of hydrophobic self-cleaning of substrate surfaces.

## Figures and Tables

**Figure 1 micromachines-14-00704-f001:**
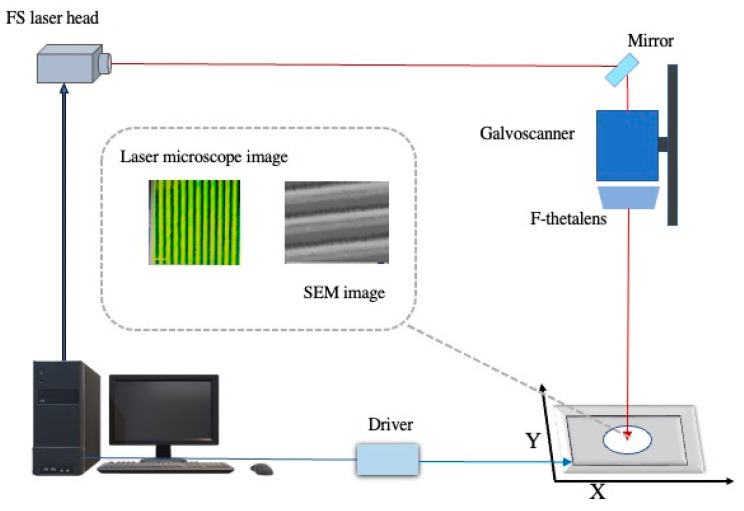
Schematic diagram of the femtosecond laser processing system.

**Figure 2 micromachines-14-00704-f002:**
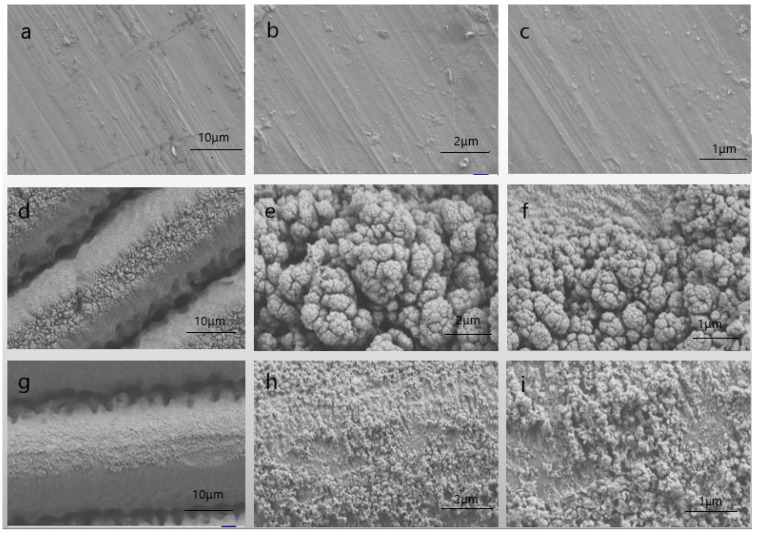
Femtosecond laser preparation of micro- and nanostructures of aluminum-based superhydrophobic surfaces. (**a**–**c**) SEM images of the surface of specimen 1#; (**d**–**f**) SEM images of the surface of specimen 2#; (**g**–**i**) SEM images of the surface of specimen 3#.

**Figure 3 micromachines-14-00704-f003:**
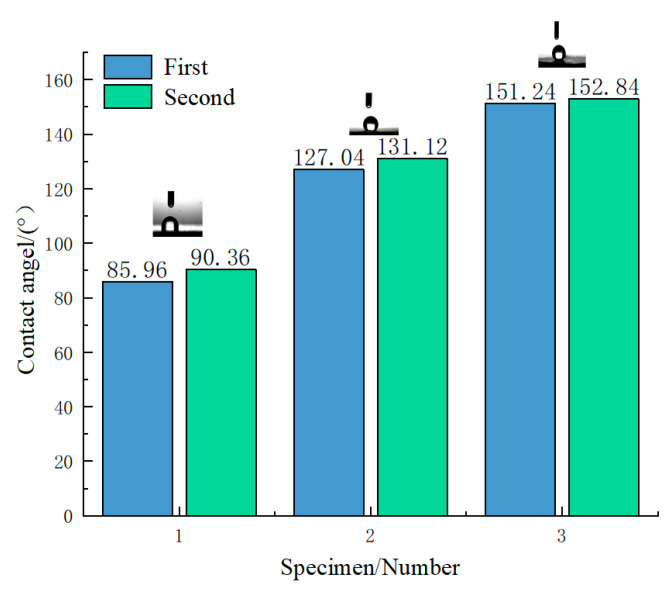
Superhydrophobic contact angle of each specimen.

**Figure 4 micromachines-14-00704-f004:**
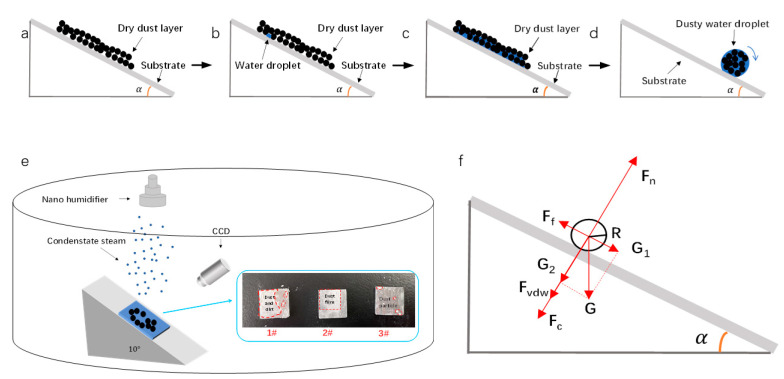
Schematic diagram of the dust removal experimental system. (**a**–**d**) Schematic diagram of the self-cleaning process; (**e**) schematic diagram of the dust removal experiment; (**f**) force analysis diagram of dust-containing droplet particles.

**Figure 5 micromachines-14-00704-f005:**
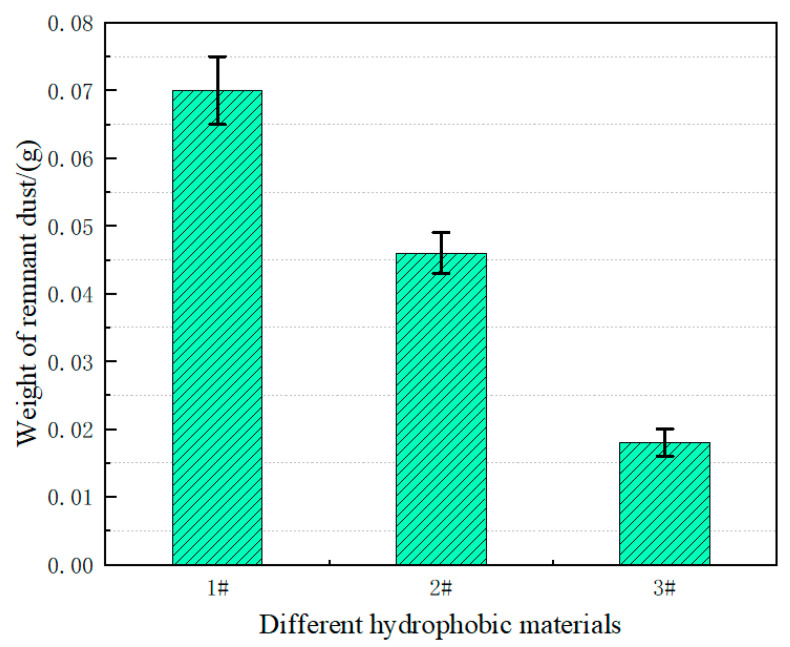
Residual dust weight of different specimens.

**Figure 6 micromachines-14-00704-f006:**
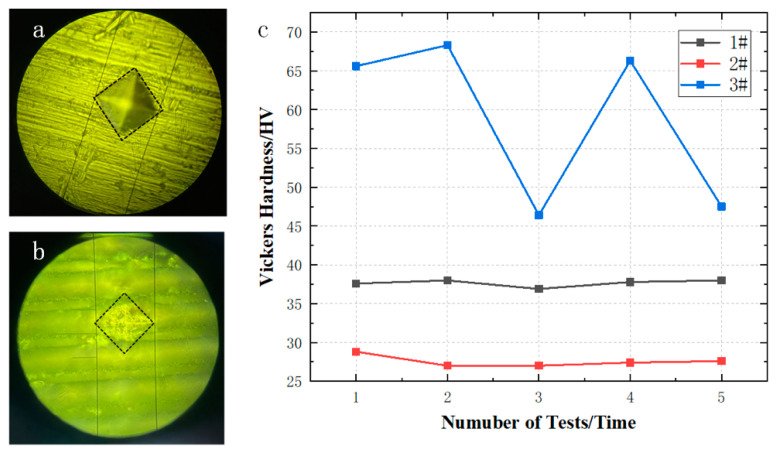
Microhardness analysis. (**a**,**b**) Optical photographs of hardness tests; (**c**) folding statistical graph of Vickers hardness for each specimen.

**Figure 7 micromachines-14-00704-f007:**
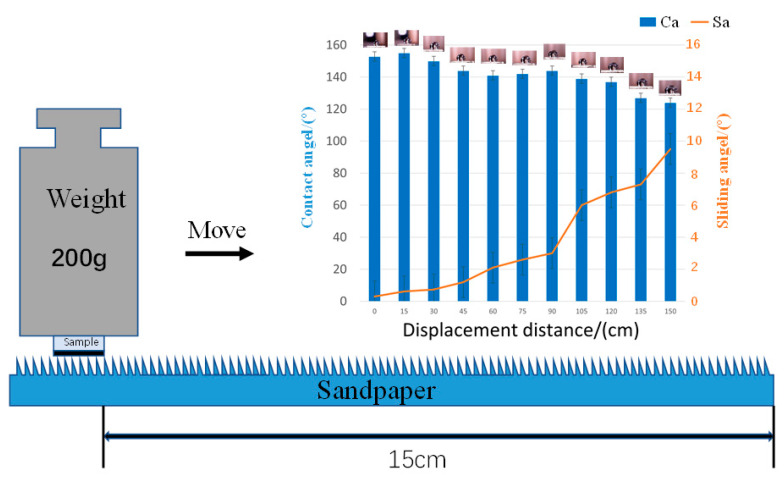
Schematic diagram of the test piece surface wear test.

**Figure 8 micromachines-14-00704-f008:**
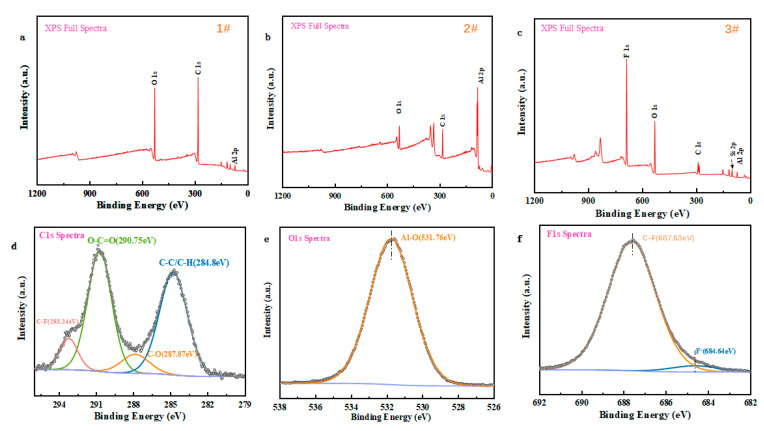
XPS spectra of each specimen. (**a**) Full spectrum of #1; (**b**) full spectrum of #2; (**c**) Full spectrum of #3; (**d**) XPS spectrum of C1s in #3; (**e**) XPS spectrum of O1s in #3; (**f**) XPS spectrum of F1s in #3.

**Table 1 micromachines-14-00704-t001:** Chemical composition of 1A99 aluminum alloy (wt.%).

Composition	Si	Al	Fe	Cu
Mass fraction/%	0.003	99.99	0.003	0.003

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
