# Peer review of "Robust, Superhydrophobic Aluminum Fins with Excellent Mechanical Durability and Self-Cleaning Ability"

_micromachines, 2023, doi:10.3390/mi14030704_

Round 1

Reviewer 1 Report

In this work, the authors prepared an aluminum base superhydrophobic surface with good mechanical stability by femtosecond laser treatment and fluorination. They also studied the surface morphology and chemical composition of their prepared samples by SEM and XPS measurements. Their experimental results show that the prepared superhydrophobic aluminum surface has excellent self-cleaning ability and excellent durability and mechanical properties. The results of this study can be useful in the field of hydrophobic self-cleaning of substrate surface. The motivation of the work and the approach adopted are well. The manuscript is publishable after the below mentioned questions/comments are addressed.

Comments:

1) The authors have not compared the results obtained in this study, including self-cleaning ability, durability and mechanical properties of the prepared superhydrophobic aluminum surface with those obtained from other methods reported in the literature.

2) What is the effect of air on the superhydrophobicity property of the prepared superhydrophobic aluminum surface?

3) How is the corrosion resistance of the prepared superhydrophobic aluminum surface?

Reviewer 2 Report

This paper shows that aluminum base superhydrophobic surface with good mechanical stability was prepared by femtosecond laser treatment and fluorination.

In the '2. Experimental Method' section, there is no detailed description of how to perform the fluorination surface treatment. (deep coating or chemical vapor deposition?)

Add the information about the specimen in the table.

What is the meaning of the first and second in Figure 3?

In Figure 6, it is necessary to explain the hardness measurement result. Why does the hardness increase rapidly after fluorination surface treatment? Can there be a large difference in hardness with chemical treatment?

Reviewer 3 Report

This article reported a robust superhydrophobic aluminum surface was obtained by modifying the surface with femtosecond laser etching and fluorination treatment. Different characterizations that conducted on modified metal surface showed excellent mechanical durability and self-cleaning ability for a broad application prospect on substrate surfaces. Comment are as follows:

1.      Suggest to put sample preparation first in the section of experimental materials and methods for these three test sample (#1 plain aluminum, #2 laser etched aluminum and #3 laser etched + fluoride treated aluminum surfaces). This way will make the article easily for readers to follow and read.

2.      How long did the fluorination being treated on the sample? How do you proceed the fluorination?  Need to mention in the experimental section.

3.      In Figure 8, XPS spectra exhibited the results of the sample #1 (plain aluminum surface) were different from #2 which was treated with femtosecond laser etching, need to provide an explanation for the difference?

4.      The article requires a sound English editing.
